# Turnover Intention and Organizational Commitment of Primary Healthcare Nurses

**DOI:** 10.3390/healthcare11040521

**Published:** 2023-02-10

**Authors:** Ana Callado, Gisela Teixeira, Pedro Lucas

**Affiliations:** 1Nursing Research, Innovation and Development Centre of Lisbon (CIDNUR), Nursing School of Lisbon, Av. Prof. Egas Moniz, 1600-096 Lisbon, Portugal; 2Amadora Health Centres, Rua Capitão Plácido Abreu, 2700-156 Amadora, Portugal

**Keywords:** nursing, primary healthcare, personnel turnover, commitment, work engagement, work environment

## Abstract

Turnover intention is a predictor of the decision to leave an organization, which, if carried out, affects the quality of care provided. There is an association between turnover intention and organizational commitment. The more committed nurses are to the unit in which they work, the more committed they become to the unit’s organizational goals; thus, they tend to continue working for the organization. Aiming to assess the turnover intention and the organizational commitment of nurses in primary healthcare, we conducted a quantitative, observational, descriptive, and cross-sectional study. The Intention of Turnover Scale and the Organizational Commitment Scale were applied in a sample of 297 nurses. Data were analyzed based on descriptive statistics. About 92.8% of the nurses intend to stay at their current workplace and only 7.3% plan to leave soon, suggesting low turnover intention; 84.5% of the nurses are willing to make an effort beyond what is normal to help their organization succeed, and 88.7% feel really interested in the destiny of the organization, which shows high organizational commitment. Pearson’s Coefficient revealed the existence of a significant negative correlation between the factors “Intention to leave” and “Committed to the organization” (r = −0.51, *p* < 0.01). These findings suggest that, when nurses are more committed to their work and to the organization, they display less intention to leave, keeping the teams committed and motivated towards the organizational goals Our findings may guide nurse managers and policy-makers to develop strategies that retain nurses in organizations, keeping them motivated and engaged, and focusing on higher organizational commitment due to the influence it may have on the turnover intention.

## 1. Introduction

Nurses’ turnover intention has a negative impact on healthcare organizations and on the nursing profession as a whole [1,2]. In 1985, Hinshaw and Atwood [3] defined turnover intention (which is considered a turnover variable) as how each person perceives the future end of their employment relationship at their current workplace. It corresponds to the worker’s willingness to end his or her contractual relationship with the organization for which he or she works [4,5,6,7] or the employee’s attempts to leave the workplace [5]. Although leaving the organization or the profession is under consideration, this intention will correspond to an internal decision process that will evolve and gain consistency over time, culminating in the materialization of the turnover [2,8,9]. It is, therefore, an excellent predictor and antecedent of the final decision to leave the organization or profession [4,9,10], reflecting the trend, mood, and starting plan.

When the willingness to leave has materialized, the costs related to the new recruitment, selection, and integration of nurses increases considerably. This is an expensive process for the organization [3]. On the other hand, the loss of experienced professionals leads to smaller, less effective, and less productive teams [11], with less ability to attend to patients’ needs and to provide high-quality care; therefore, the departure of experienced nurses represents a loss of value for the organization [3]. This process often begins with a change of unit within the organization, then the organization itself; finally, the nurse leaves the profession entirely [2]. This suggests that the service and the organization play an important role in this intention [9]; thus, this concept is particularly important for management, as being aware of this intention provides managers with time to plan and intervene in the actual turnover [3].

Scientific evidence supports the existence of a significant positive relationship between turnover intention and organizational commitment [1,4,5,9,10]. Nurses’ organizational commitment is considered crucial to the goals of healthcare organizations since it relates to the professional’s relationship with the organization for which he or she works [12]. It is, therefore, understood as a dimension of organizational efficiency [7].

Organizational commitment has been defined as the relative strength of an individual’s identification and involvement with a particular organization, characterized by a strong belief in its goals and values, a willingness to exert considerable effort on behalf of the organization, and a strong desire to continue working for the organization in question [13]; therefore, it refers to an individual’s connection and identification with the organization and their loyalty to it, manifested through involvement in the organization’s activities [4,10,13]. The more committed nurses are to the unit in which they work, the more likely they are to engage in organizational activities and goals [13]. This helps to improve patients’ satisfaction, safety, and quality of care, leading to higher levels of job satisfaction [14] and reduced turnover [4]; thus, organizational commitment is predictive of the workers’ behavior regarding their decision to stay with or to leave the organization [4,5,7,10].

Nurse managers’ intervention in the performance of the nursing team is essential, influencing the turnover intention by creating positive or negative feelings in nurses [2,3,15], subsequently increasing their organizational commitment and job satisfaction and reducing turnover [14]. Through specific health-management skills and knowledge, as well as leadership skills, the nurse manager should promote favorable nursing work environments [3,16,17,18]. A favorable nursing work environment is essential to the success of healthcare systems and is related to the quality of nursing care, job satisfaction, retention of good professionals, patient safety, and effective care, as well as the efficiency of organizations [16,17,18,19,20,21,22,23].

The growing demand for nursing care and the increased complexity in primary healthcare delivery caused by global aging [24], the transfer of chronically ill patients to the community—home, schools, etc.—to be monitored by their teams [25], the reduced number of nurses practicing in this context [24], and the challenges arising from the ongoing COVID-19 pandemic require the efforts of motivated and committed professionals to meet organizations’ goals [16,25]. For this reason, it is important for nurse managers to be aware of the turnover intention and organizational commitment of their nurses in order to implement the necessary strategies and changes to keep their teams motivated and committed to the organization’s goals, maintaining a high-quality practice [26] and keeping their teams stable.

From previous research, we observed this theme has been less studied in primary healthcare settings. We verified that no studies including these variables have been conducted in primary healthcare in Portugal; thus, there is a knowledge gap that our study can help to fill.

## 2. Materials and Methods

### 2.1. Aim

This study aimed to analyze the relationship between the turnover intention and the organizational commitment of nurses in primary healthcare.

### 2.2. Study Design

A quantitative, observational, descriptive, and cross-sectional study was conducted.

### 2.3. Participants

The study sample consisted of 297 nurses working in a primary healthcare center in the Lisbon region, in Portugal. Considering that the total population comprised 345 nurses, our sample corresponds to a response rate of 86.1%.

As inclusion criteria, we defined nurses working in primary healthcare settings.

### 2.4. Data Collection

Data were collected between December 2019 and January 2020. The data collection instrument was composed of sociodemographic variables: age, gender, nursing professional experience, and academic qualifications; by the Intention of Turnover Scale of De Sul & Lucas (2020); and by the Organizational Commitment Scale of Gomes (2007). Both scales were validated for the Portuguese cultural context.

The Intention of Turnover Scale of De Sul & Lucas [3] was translated and validated for the Portuguese cultural context in 2020, from the original Anticipated Turnover Scale of Hinshaw and Atwood, which was developed in 1984 [3]. It is a Likert-type scale with 10 items, rated from 1 (strongly disagree) to 7 (strongly agree). It can achieve a maximum of 70 points and a minimum of 10 points. The higher the score, the higher the nurses’ turnover intention. This scale has been widely used in international studies [4,8]. It presents high internal consistency, with a Cronbach’s alpha value of 0.94 [27].

The Organizational Commitment Scale was translated and validated for the Portuguese cultural context by Gomes in 2007, adapted from the original Organizational Commitment Questionnaire developed in 1979 by Mowday, Porter, and Steers [28]. Gomes’ Organizational Commitment Scale is composed of nine items and assesses professionals’ feelings, attitudes, and positive values related to their workplace. It is another Likert-type scale, rating from 1 (totally disagree) to 5 (totally agree). The total score should be summed and divided by the number of items. High scores correspond to high levels of organizational commitment [4]. This scale is still widely used in international studies [4,29], since it has achieved high levels of reliability and high internal consistency, with a Cronbach’s alpha value of 0.90 [4].

### 2.5. Ethical Considerations

This study was approved by the Ethics Committee of the organization. Nurses asked to participate provided their informed consent, having received detailed information about the study. Their participation was voluntary, and the participants had the option to withdraw from the study at any time. Confidentiality and anonymity were guaranteed.

### 2.6. Data Analysis

A descriptive and comparative analysis of variables was performed. Continuous variables were described using descriptive statistics, such as the mean and standard deviation. Categorical variables were described using the relative and absolute frequencies. We used Pearson’s Coefficient to analyze the relationship between turnover intention and organizational commitment.

The statistical treatment of data was performed using the statistical software IBM-SPSS Statistics, version 26.0 (v.26, SPSS, Inc., Chicago, IL, USA).

## 3. Results

### 3.1. Sociodemographic Characterization of the Sample

Our sample was composed mostly of women (91.8%); the average age was 48 years; participants were predominantly married or cohabiting (78.4%) (Table 1). Most nurses had a post-graduate education—including specialist, post-graduate, and master’s degrees (60.8%)—and fit the professional category of nurse specialist (52.6%). The average time participants had worked in the profession was 24.8 years (SD = 7.7) and the median time they had spent in their current workplace was 6.0 years (SD = 5.4) (Table 1).

Despite being mainly composed of women and nurse specialists, our sample does not differ from the samples of other studies, where the percentage of women was higher than 90%, and between 70% and 90% of the participants had post-graduate education [2,7,30,31,32].

### 3.2. Turnover Intention and Organizational Commitment

Regarding the turnover intention, we verified that 92.8% of nurses agreed with the item “1. I intend to stay at my current…” and only 13.3% agreed with the item “2. I am pretty sure I will leave…” (Table 2). The mean scores of the Turnover Intention Scale (2.8, SD = 0.90) and of the factor “intention to leave” (3.1, s = 1.0) were lower than the cutoff value (3.5), showing that there was no turnover intention in our sample. The mean score of the factor “intention to stay” (6.0, SD = 1.2) was higher than the cutoff value, which supports the previous results (Table 3).

Regarding the sociodemographic variables, we verified that, with the exception of the variable length of service in the current functional unit, the mean was lower than the cutoff value (3.5), reaching the highest mean values in single men aged 32–35 years, with the professional category of nurse manager, nurses with post-graduate education, working in the Public Health Unit, working in the profession for 10–18 years, and working in the current functional unit for 17–32 years. Despite the mean differences mentioned above, notable significance levels were only found in the sociodemographic variables “academic education” and “current functional unit” with *p*-values < 0.05; thus, for the remaining sociodemographic variables, we concluded that there is homogeneity of variances in turnover intention.

With reference to organizational commitment, we found that 84.6% of the participants agreed with the item “11. I am willing to make an effort beyond what is normal…”. The analysis of the remaining items allows us to conclude that nurses in our study are committed to their organization (Table 2). The mean score of the Organizational Commitment Scale, which corresponds to the factor “committed to the organization” (3.7, s = 0.7), being higher than the cutoff value (2.5), allows us to conclude that participants feel very committed to the organization for which they work (Table 3). By analyzing the mean values obtained in the sociodemographic variables, we found that organizational commitment is higher in men, in widows/those who are separated, those aged between 52–55 years, with the professional category of nurse and with nursing diploma, in those working in the community care unit, and those who have been working in the profession for 27–34 years and in their current functional unit for 9–16 years. Despite these differences, we concluded that there are only statistically significant differences in the organizational commitment for the following sociodemographic variables: current functional unit and length of service in the current functional unit (*p* < 0.05). The Bonferroni post hoc test identified significant differences in the organizational commitment of nurses working in the community care unit compared to those working in the personalized healthcare unit and between those working in the family healthcare unit compared to those working in the personalized healthcare unit. Significant differences were also identified in nurses who worked in the current functional unit for 9–16 years compared to those who worked in the current functional unit for 1–8 years. For the remaining sociodemographic variables, we concluded that there was homogeneity of variances.

### 3.3. Relationship between Turnover Intention and Organizational Commitment

The Pearson’s Coefficient allowed us to analyze the relationship between the turnover intention and organizational commitment. When correlating turnover intention with organizational commitment, we obtained the value of r = −0.54 (*p* < 00.1), which shows that there is a significant negative correlation between these variables. Pearson’s Coefficient also revealed the existence of a significant negative correlation between the factors “intention to leave” and “committed to the organization” (r = −0.51, *p* < 0.01); however, for the factors “intention to stay” and “committed to the organization”, we found a significant positive correlation (r = 0.39, *p* < 0.01), meaning that when organizational commitment increases or decreases, the intention to stay varies in the same direction; thus, we found that there is a significant negative correlation between turnover intention and organizational commitment.

## 4. Discussion

### 4.1. Nurses’ Turnover Intention in Primary Healthcare

The results indicate that nurses in our sample have no intention of turnover. We found that only 13.2% intended to experience turnover soon. These findings differ from Aiken et al. [11], whose study was conducted in 12 European countries, in hospital settings, with the purpose of identifying nurses’ perceptions of the nursing work environment, and the quality of nursing care, where it was found that 20% to 40% of nurses in all countries intended to leave their current workplace in the next year.

Additionally, in the study conducted by Almalki et al. [1] in primary healthcare settings, aiming to correlate nurses’ quality of life with their turnover intention in Saudi Arabia, the authors found that 40.4% of the nurses expressed an intention to leave their current workplace. Finally, in the study conducted by Yurumezoglu et al. [2], which aimed to assess the use of evidence in nursing management, 70.4% of the nurses expressed their intention to leave their current organization.

Despite not presenting turnover intention, our sample shows higher scores in men, single people, and those in the younger age group (32–35 years), as well as those of a higher professional category, those who have worked in the profession for 10–18 years, and those who have worked in their current functional unit for 17–32 years; however, no statistical significance was found in these groups. We obtained higher values of turnover intention, and with statistical significance, in nurses with higher academic qualifications and those working in the public health unit.

These results are in line with the scientific evidence. Studies that obtained significant differences in the academic education/professional sociodemographic variables concluded that nurses with higher levels of education seek a practice that allows them to apply their advanced knowledge and they also seek recognition for their skills [1]. On the other hand, workload, lack of autonomy, performance of tasks unrelated to nursing practice, and limited time to perform the activities, among others, may be some of the factors that increase the turnover intention [1,11,33].

Our findings may be related to several factors. About 78% of the nurses in our sample were very happy to have chosen to work in primary healthcare. This choice may be related to the recognized importance of primary healthcare, which is considered a key to achieve the level of health that allows the whole population to live an economically and socially productive life, and whose main function is through the collaborative work of several professionals [34] to promote health and prevent disease. This allows widening of the nurses’ field of action and their professional development [35]. The specific organization of nursing activities in the several units, by healthcare teams, by healthcare programs, or in health promotion projects targeting groups, may promote the autonomy of nurses, leading to increased motivation and greater willingness to remain in the organization [33]. Professional autonomy is considered a vital element for nurses and for a favorable nursing work environment [36]; on the other hand, the working hours have an impact on nurses’ willingness to stay or leave an organization, with normal working hours being associated with a lower turnover intention [37]. In primary healthcare, the working hours are between 8 am and 8 pm, allowing for greater compatibility with personal needs. Finally, studies have identified that lower salaries are associated with higher turnover intention [1]. In Portugal, the recent change in the nursing career, which included the category of nurse specialist, allowed several nurses to make the transition to this new category and obtain a pay increase, which may justify a greater intention to stay in the organization.

### 4.2. Organizational Commitment of Nurses in Primary Healthcare

The nurses in our study feel committed to their organization. The majority of nurses are willing to make an effort for the organization (84.5%) and feel their personal values coincide with the organizational values (71.1%).

In our sample, the mean value of organizational commitment was higher in men, in participants with a widow/separate marital status, in the age group 52–55 years, in participants with the professional category of nurse, and in those with nursing diploma, who have been practicing for 27–34 years, although no significant differences were found between groups in these sociodemographic variables.

We verified a statistical significance in the following sociodemographic variables: functional unit of current exercise and length of service in the current functional unit; thus, participants working in the community care unit and working in the current functional unit for 9–16 years had higher levels of organizational commitment.

Several studies found that, among other sociodemographic characteristics, the length of service in the organization influences organizational commitment [6]; on the other hand, conflictive nursing work environments with role ambiguity constitute the organizational factors with the greatest influence on the levels of organizational commitment [4]. In this vein, it has been found that the context of care practice has a great influence on job satisfaction, personal motivation, stress, and burnout, which influence attachment, identification, and willingness to make a considerable effort for the organization and its objectives, making it more efficient [5]; on the other hand, the length of time spent working in the same functional unit is probably related to higher levels of job satisfaction and greater identification with organizational values, as well as the existence of correspondence between personal expectations and organizational objectives [38].

The fact that the nurses in our sample have chosen to work in primary healthcare—some of them for several years, others more recently—may be due to certain characteristics associated with this context [6]. In primary healthcare, the organization of small, multi-professional teams with similar objectives stimulates collaboration and mutual help, which is fundamental for the achievement of goals. This teamwork towards a common goal often requires the performance of complementary activities and mutual support, benefiting the organization and positively stimulating organizational commitment [39]. The creation of small teams stimulates the internal organization, leading to a higher degree of autonomy. This circumstance may provide the perception of support, participation in decision-making, and professional autonomy; these variables are widely associated with higher levels of job satisfaction, better performance, and higher levels of organizational commitment among nurses [36]. The cooperation and support of the team, as well as the accomplishment of interesting tasks, increase the perception of identification with the organizational objectives, the energy devoted to the activities performed, and the development of positive feelings, motivating professionals to stay with the organization [39].

### 4.3. Relationship between Nurses’ Turnover Intention and Organizational Commitment in Primary Healthcare

Our findings of the significant negative correlation between turnover intention and organizational commitment are similar to those found in several international studies [4,37].

In a study conducted by Han et al. [4], which aimed to determine the effects of a work environment and role ambiguity on turnover intention, and the mediating effects of organizational commitment and burnout, the results highlighted the negative correlation established between organizational commitment and turnover intention, since it was found that nurses with higher levels of organizational commitment had lower turnover intention; similarly, Brunetto et al. [37], who aimed to assess the impact of psychological capital and support by managers on turnover intention, identified the contributions of job satisfaction, organizational commitment, and stress on turnover intention. In this study, the existence of a negative correlation between organizational commitment and turnover intention was also verified.

The work experience related to the existing characteristics of the organization, the co-workers, the managers, and the responsibilities have an impact on the levels of organizational commitment [9,40]. The more committed the employee is to work and the organization, the less he/she wants to leave the workplace [10].

Our results allow us to direct nurse managers and policy-makers to develop strategies to retain motivated and committed nurses in organizations, due to the role that organizational commitment plays in turnover intention. Nurse managers are a key element due to their role in nursing work environments and their ability to influence the organizational commitment and turnover intention of the nursing team. They should promote better working conditions through the reorganization of the services [26] and the implementation of conceptual models of clinical practice that allow patients’ needs to be met [17]. Performance management strategies are also highlighted [26]; these include access to professional development via formal education and feedback [11,14], as well as the opportunity to engage in interesting activities, stimulating autonomy and participation in decision-making, and, finally, providing support for a high-quality clinical practice [11].

### 4.4. Limitations

This study was conducted shortly after the transition to a new nursing career, which may have caused an increase in the perception of organizational appreciation and recognition. This feeling may have influenced the participants’ answers; thus, further studies should be conducted to compare the results.

## 5. Conclusions

The nurses who participated in our study have low turnover intention and high organizational commitment. We found a significant negative relationship between turnover intention and organizational commitment, which highlights the importance of promoting high levels of organizational commitment as a strategy to retain the best nurses and keep them committed to the organization’s goals. Our results may contribute to informed decision-making by nurse managers and policy-makers.

Despite the recognized importance of primary healthcare, there are few international studies exploring this topic in a healthcare setting. To our knowledge, this is the first study of this nature to be carried out in Portugal. Further studies are needed before the conclusions can be generalized.

## Figures and Tables

**Table 1 healthcare-11-00521-t001:** Sociodemographic Characterization of the Sample.

Sociodemographic Variables	%	Mean	Median	Standard Deviation
Gender				
Male	8.25			
Female	91.75			
Marital Status				
Single	13.4			
Married/Cohabiting	78.4			
Widow/Separated	8.2			
Age Groups				
32–35	5.2	47.9	48.0	7.8
36–39	15.5			
40–43	10.3			
44–47	15.5			
48–51	16.5			
52–55	18.6			
56–59	14.4			
60–63	4.0			
Professional Category				
Nurse	42.3			
Nurse Specialist	52.5			
Nurse Manager	5.2			
Academic Qualifications				
Nursing Diploma	39.2			
Post-graduate Education	60.8			
Current Functional Unit (FU)				
Community Care Unit	20.6			
Personalized Healthcare Unit	23.7			
Family Healthcare Unit	48.5			
Public Health Unit	5.2			
Other	2.0			
Years of Professional Activity				
10–18	23.6	24.8	25.0	7.7
19–26	38.2			
27–34	26.8			
35–42	11.4			
Length of Service in the Current FU (years)				
1–8	68.0	-	6.0	5.4
9–16	24.8			
17–32	7.2			

**Table 2 healthcare-11-00521-t002:** Results obtained by applying the Intention of Turnover Scale and the Organizational Commitment Scale.

Intention of Turnover Scale —De Sul and Lucas (2020)	Item Concordance		
	Totally Disagree	Without Opinion	Totally Agree
1. I intend to stay at my current workplace for some time.	6.2%	1.0%	92.8%
2. I am pretty sure I will leave my workplace in the near future.	81.5%	5.2%	13.2%
3. Deciding to stay or leave my workplace is not a key issue for me at the moment.	69.1%	16.5%	14.5%
4. If I received another job offer tomorrow, I would seriously consider it.	39.1%	28.9%	31.9%
5. I have no intention of leaving my current workplace.	19.5%	11.3%	69.1%
6. I’ve been in this workplace as long as I wanted to.	58.8%	19.6%	21.6%
7. I’m sure I will be here for some time.	8.3%	7.2%	84.6%
8. I intend to keep my job in this organization for some time.	6.1%	5.2%	88.7%
9. I have serious doubts about whether or not I will actually stay in this organization.	68%	12.4%	16.6%
10. I plan to leave this workplace soon.	84.5%	8.2%	7.3%
Organizational Commitment Scale—Gomes (2007)			
11. I am willing to make an effort beyond what is normal to help this organization succeed.	4.1%	11.3%	84.5%
12. I tell my friends that this organization is a great place to work.	9.2%	21.6%	69.1%
13. I am willing to accept almost any kind of assignment so that I can continue to work in this organization.	61.8%	19.6%	18.6%
14. I find that my personal values and those of this organization are quite similar.	17.5%	27.8%	54.6%
15. I am proud to tell others that I work in this organization.	6.2%	22.7%	71.1%
16. I feel inspired to do my best by the fact that I work in this organization.	3.1%	14.4%	82.4%
17. I feel very happy to have chosen this organization to work in.	4.1%	17.5%	78.4%
18. I am really interested in the destiny of this organization.	3.1%	8.2%	88.7%
19. For me, this is the best place to work.	12.4%	34%	53.6%

**Table 3 healthcare-11-00521-t003:** Scores obtained in the Intention of Turnover Scale, in the Organizational Commitment Scale, and in the three latent factors.

	Mean *^/^**	Standard Deviation
Intention of Turnover Scale ^1,^*	2.8	0.9
Factor “Intention to Leave” ^1,^*	3.0	1.0
Factor “Intention to Stay” ^2,^*	6.0	1.2
Organization Commitment Scale ^2,^** and “Committed to the Organization” Factor ^2,^**	4.0	0.7

* Cutoff = 3.5; ** cutoff = 2.5; ^1^—lower the better; ^2^—higher the better.

## Data Availability

Not applicable.

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
