# Peer review of "Turnover Intention and Organizational Commitment of Primary Healthcare Nurses"

_healthcare, 2023, doi:10.3390/healthcare11040521_

Round 1

Reviewer 1 Report

Table 1 unreadable 

Table 2 illegible 

Table 3 illegible 

 Proposals for change

Summary: to change

Title to be changed

No DOI number

Author Response

The manuscript has extensive editing of the language and English style done.

Reviewer 2 Report

Overall, it is essential that your manuscript needs editing or proofreading. A list of your references needs to follow the guideline of the journal in the following link 

https://www.mdpi.com/journal/healthcare/instructions

 In the abstract, you need to add one or more Sentences about the result of your study and a conclusion of your studies. 

The introduction needs to include more studies that examine your topic." Turnover intention and organizational commitment of primary health care nurses."

I could not find the gap in the study; can you please add the gap? (The last part of your discussion needs to move to the end of the introduction as a gap of knowledge. )

Methodology: 

  • A sampling of the study: one of the significant comments in the paper, there is no information population, sampling size, generalization, response rate, inclusion-exclusion criteria, repose rata, or response rate....etc.
  • You need to add more information about the setting of your study. 

Discussion 

 There is a need for further explanation of your study's recommendations. 

The limitation of your studies needs to rewrite and improved it. 

Author Response

(The authors gave the same response as above.)

Reviewer 3 Report

Pedro Lucas et al. submitted to Healthcare an article on turnover intention and organizational commitment of primary healthcare nurses.

Although I was hesitant evaluating a study in which “Data were collected between December 2019 and January 2020”, three years ago and therefore apparently obsolete, this manuscript is supported by fairly recent references, which nonetheless corroborate its discussions and conclusions.

The manuscript is well structured, it has a good scientific soundness and, with the appropriate changes and implementation, it could be favorably re-evaluated for its eventual acceptance.

Here are my considerations:

L87: “The study sample consisted of 297 nurses working”: well, the sample consists of 297 enrolled nurses, but what is the general reference population (denominator)? Using a simple size calculator, are the criteria and sufficient numbers met to apply statistical inference? In other words, is the sample investigated representative of the entire population?

- was a pilot study previously conducted, useful for evaluating the intelligibility of the questionnaire? 

- Table 1 -> section Professional Category: please be careful, because the sum of the percentage values is not 100. I believe it is important to meticulously reassess if the sum of the numerical values declared in all the tables is compliant.

LL 228-229 or in LL268-270: please evaluate the opportunity to complete the sentence, incorporating the concept of multi-professionalism in which the Preventive Health Professions also find their place in these operating lines, working together with nurses in primary healthcare and health promotion settings: DOI: 10.3390/healthcare10101906

- in the discussions, please evaluate the opportunity to also consider the contents of the local sector literature on the topic under examination, assessing the following manuscript: DOI: 10.1177/1744987118789009 and DOI: 10.1590/0034-7167-2020-0782

- According to the Authors, what are the strategies that Nurse Managers can adopt to fight turnover? Are there actions, methods, organizational contexts and strategies to promote an increasingly engaging and proactive organizational mood? By briefly developing these aspects, a scientific work that could be published in a Journal such as Healthcare is even more attractive.

Thank you for your efforts in perfecting this important article!

Author Response

(The authors gave the same response as above.)

Round 2

Reviewer 2 Report

Point 1. Overall, it is essential that your manuscript needs editing or proofreading. A list of your references needs to follow the guideline of the journal in the following link  https://www.mdpi.com/journal/healthcare/instructions

There is improvement in the documents, however, still, you need to review the journal reference guild line and make sure that fallow the journal guidelines 100%.

Point 2. In the abstract, you need to add one or more Sentences about the result of your study and a conclusion of your studies. 

I think you need a summary of all major findings in the result, not just one finding.

Thank you 

Author Response

Dear reviewer,

Thank you for the precious and valuable comments/recommendations for improvement. Kindly check below our responses to it.

Point 1. Overall, it is essential that your manuscript needs editing or proofreading. A list of your references needs to follow the guideline of the journal in the following link  https://www.mdpi.com/journal/healthcare/instructions

There is improvement in the documents, however, still, you need to review the journal reference guild line and make sure that fallow the journal guidelines 100%.

Response 1: Thank you for your comment. As you can see, all the references have been rectified and the DOI has been added, when it exists.

Point 2. In the abstract, you need to add one or more Sentences about the result of your study and a conclusion of your studies. 

I think you need a summary of all major findings in the result, not just one finding.

Response 2: Thank you for your comment, which considerably improved our work. As you can see, we have added more information about the study results in the abstract.

We take this opportunity to inform that this version of the manuscript has already been subject to English editing services from MDPI.

Thank you very much!!!

Reviewer 3 Report

The Authors have provided exhaustive answers and modified their work, taking into account the suggestions provided. Using a simple size calculator, the minimum number of enrolled nurses for the application of statistical inference is met. The sums of the values included in the tables are compliant. Furthermore, the manuscript has been completely revised regarding the English syntax and form. Thank you for your commitment and meticulousness.

Author Response

Dear reviewer,

Thank you for the precious and valuable comments/recommendations for improvement. Kindly check below our responses to it.

Point 1. The Authors have provided exhaustive answers and modified their work, taking into account the suggestions provided. Using a simple size calculator, the minimum number of enrolled nurses for the application of statistical inference is met. The sums of the values included in the tables are compliant. Furthermore, the manuscript has been completely revised regarding the English syntax and form. Thank you for your commitment and meticulousness.

Response 1: Thanks for your comment and such positive feedback.

We take this opportunity to inform that this version of the manuscript has already been subject to English editing services from MDPI.

Thank you very much!!!
